# Adverse Maternal and Infant Outcomes of Women Who Differ in Smoking Status: E-Cigarette and Tobacco Cigarette Users

**DOI:** 10.3390/ijerph20032632

**Published:** 2023-02-01

**Authors:** Eline K. Nanninga, Stella Weiland, Marjolein Y. Berger, Esther I. Feijen-de Jong, Jan Jaap H. M. Erwich, Lilian L. Peters

**Affiliations:** 1Amsterdam UMC, Location Vrije Universiteit Amsterdam, Midwifery Science, De Boelelaan 1117, 1081 HV Amsterdam, The Netherlands; 2Midwifery Academy Amsterdam Groningen, InHolland, Dirk Huizingastraat 3-5, 9713 GL Groningen, The Netherlands; 3Department of General Practice & Elderly Care Medicine, University Medical Centre Groningen, University of Groningen, Hanzeplein 1, 9713 GZ Groningen, The Netherlands; 4Amsterdam Reproduction and Development, De Boelelaan 1117, 1081 HV Amsterdam, The Netherlands; 5Department of Obstetrics & Gynecology, University Medical Centre Groningen, University of Groningen, Hanzeplein 1, 9713 GZ Groningen, The Netherlands

**Keywords:** prenatal exposure, electronic cigarettes, health consequences, nicotine, derivatives

## Abstract

The electronic cigarette (e-cigarette) became commercially available around 2004, yet the characteristics of pregnant women who use these devices and their effects on maternal and infant health remain largely unknown. This study aimed to investigate maternal characteristics and pregnancy outcomes according to maternal smoking status. We conducted a cross-sectional study of Dutch women with reported pregnancies between February 2019 and May 2022, using an online questionnaire to collect data on smoking status and demographic, lifestyle, pregnancy, and infant characteristics. Smoking status is compared among non-smokers, tobacco cigarette users, e-cigarette users, and dual users (tobacco and e-cigarette). We report descriptive statistics and calculate differences in smoking status between women with the chi-square or Fisher (Freeman–Halton) test. Of the 1937 included women, 88.1% were non-smokers, 10.8% were tobacco cigarette users, 0.5% were e-cigarette users, and 0.6% were dual users. Compared with tobacco users, e-cigarette users more often reported higher education, having a partner, primiparity, and miscarriages. Notably, women who used e-cigarettes more often had small infants for gestational age. Despite including few women in the e-cigarette subgroup, these exploratory results indicate the need for more research to examine the impact of e-cigarettes on pregnancy outcomes.

## 1. Introduction

Smoking tobacco cigarettes during pregnancy is associated with low birth weight, preterm birth, neural underdevelopment, and stillbirth [1,2,3,4]. The mechanisms behind these effects are largely attributable to nicotine and carbon monoxide released by burning tobacco, which may decrease placental blood flow and contract fetal arteries [3,5]. Tobacco cigarettes also affect the health of the mother, being associated with gestational hypertension, preeclampsia, gestational diabetes, and postpartum hemorrhage [6,7,8].

The electronic cigarette (e-cigarette) was introduced in 2004 as a healthier alternative to regular tobacco cigarettes for heavy smokers [9,10] with the main use scenario being to facilitate smoking cessation [11]. They benefit from feeling like conventional cigarettes to use, having the ability to add pleasant flavors, and heating an "e-liquid" instead of burning tobacco [12]. However, e-liquids cannot be considered safe because they can still contain nicotine (0–36.6 mg/mL) and other potentially harmful chemicals with known adverse health effects in non-pregnant users and a growing prevalence of e-cig/vaping-associated lung injury (EVALI) in the United States [10,13,14,15]. EVALI is characterized by dyspnea, cough, and hypoxemia with bilateral airspace opacities on chest imaging, where patients often have to be admitted to the intensive care unit [15]. The continued growth in their popularity, especially among teenagers, is a cause for concern [16]. As of 2021, about 1.4% of adults in the Netherlands used e-cigarettes regularly, with an estimated 0.4% of pregnant women reportedly using substances like hookah, nitrous oxide, and/or e-cigarettes in 2018 [17,18]. Given that e-cigarettes are not benign devices, often still contain nicotine, and appear to show increased use among women of childbearing age, we urgently require a better understanding of the risks to both the mother and infant.

Studies of how e-cigarettes affect maternal and infant outcomes have produced inconclusive results to date, with most focusing on nicotine exposure and its effects on blood flow [6,19,20]. Consistent with the literature on tobacco smoking, data from animal and laboratory research have shown that nicotine may cause poor placentation, while data from animal studies and one cohort study suggest it may cause low birthweight [21,22,23,24]. However, in a review of fetal toxicity associated with e-cigarette use, Greene and Pisano concluded that we still lack strong epidemiological evidence from studies comparing e-cigarette use with either tobacco smoking or non-smoking during pregnancy [25]. To our knowledge, only one questionnaire-based observational study has compared the characteristics of pregnant women who used e-cigarettes to those of pregnant women who used tobacco cigarettes or did not smoke. Of the 4442 British women included in that research, the 2.8% who used e-cigarettes during pregnancy were mostly younger, of British origin, lived in deprived areas, had left full-time education at a younger age (≤18 years), and had a partner who also smoked when compared with non-smokers and tobacco users [26]. There is no comparable research in the Dutch population, which must be corrected to know where to target research and public health initiatives.

Existing guidance on e-cigarette use during pregnancy largely relies on evidence from studies into either tobacco cigarette smoking during pregnancy or the chemicals and toxins in e-cigarette smoke combined. This lack of specific evidence led the Dutch Association of Obstetricians and Gynecologists (NVOG) and the Royal Dutch Association of Midwives (KNOV) to recommend discouraging e-cigarette use during pregnancy [27]. Unraveling the complex relationship between e-cigarette use and maternal and infant outcomes will inform future iterations of this guidance and the advice given to pregnant women. An important first step is to define the characteristics of women who use e-cigarettes during pregnancy.

This study aimed to investigate the individual characteristics and adverse maternal and infant outcomes of women according to their smoking status, comparing non-smokers, tobacco users, e-cigarette users, and dual users of tobacco and e-cigarettes.

## 2. Materials and Methods

### 2.1. Study Design and Ethics

This cross-sectional study used data collected by an online questionnaire between March 2020 and May 2022. We targeted Dutch-speaking women aged ≥ 16 if they had been pregnant between February 2019 and May 2022 and lived in the Netherlands. The questionnaire was completed at most one year after birth. Women were excluded if they did not consent to the anonymous use of their data or if they had missing data for any inclusion criterion and/or their smoking status (i.e., non-smokers, tobacco cigarette users, e-cigarette users, or dual users). This type of research does not require ethical approval in the Netherlands, and the ethical review board of the University Medical Hospital Groningen provided a waiver stating that the Medical Research Involving Human Subjects Act (WMO) does not apply (number: METc 2019/099).

### 2.2. Recruitment Strategy

Women received an invitation to complete the online questionnaire via four routes: (1) 456 primary midwifery care practices in the Netherlands; (2) social media posts by Midwifery Academy Groningen Amsterdam and participating researchers (e.g., personal LinkedIn pages); (3) posts on 3 forums, 37 Facebook groups, and 5 Facebook pages related to pregnancy and birth; and (4) targeting e-cigarette users through the municipal health services of Groningen, Friesland, and Drenthe. Completing the questionnaire was voluntary and was not rewarded with any compensation.

### 2.3. Questionnaire Construction

The online questionnaire comprised 90 questions, of which 22 were open-ended and 68 were closed, with data collection performed using Google Forms. Depending on the women’s answers for smoking status and end of pregnancy, they answered different questions targeted at their specific situation. This study collected the following data on demographic and lifestyle characteristics: maternal age, migration background (i.e., Western or non-Western), education level (i.e., low, middle, or high), marital status (no partner or partner), smoking status (i.e., non-smokers, tobacco cigarette users, e-cigarette users, or dual users), smoking duration, e-cigarette nicotine dose (i.e., none, low [≤10 mg], medium [11–22 mg], or high [≥23 mg]), previous smoking, second-hand smoke exposure, and pre-pregnancy body mass index (kg/m^2^). We also collected data about pregnancy characteristics (i.e., mode of conception, parity), maternal outcomes (mode of birth, hypertensive disorders of pregnancy, gestational diabetes, or postpartum hemorrhage >1 L), and infant outcomes (e.g., birth weight, gestational age, size for gestational age, hospital admission within 1 year, or perinatal death). Adverse outcomes associated with smoking tobacco cigarettes were included on the basis that they might also be associated with e-cigarette use [1,2,4,6,7,8,28].

### 2.4. Statistical Analysis

Demographic, lifestyle, and pregnancy characteristics, together with maternal and infant outcomes, are reported descriptively and stratified by smoking status. Adverse maternal and infant outcomes are grouped as composite dichotomous variables, with their presence operationalized as having at least one adverse outcome. For adverse maternal outcomes, we considered hypertensive disorders of pregnancy, gestational diabetes, and/or postpartum hemorrhage, whereas for adverse infant outcomes, we considered preterm birth, small for gestational age (SGA), hospital admission, and/or stillbirth. Missing data on items of the questionnaire was reported; women were not excluded from the missing values. Statistical differences in the characteristics and outcomes were calculated by smoking status using chi-square or Fisher (Freeman–Halton) exact tests, as appropriate. The Monte Carlo test was used if the Fisher exact test could not be calculated. A *p*-value of ≤0.05 was defined as statistically significant, and all analyses were performed with IBM SPSS Version 25.0 (IBM Corp., Armonk, NY, USA).

## 3. Results

### 3.1. Participants and Descriptive Data

In total, 2041 women completed the questionnaire, from which we excluded 81 for not meeting the eligibility criteria (Figure 1). We also excluded 23 who completed the online questionnaire more than once, which probably occurred due to technical problems (e.g., failure to load the next page or internet connectivity issues). Missing values on items of the questionnaire ranged from 0% (items about maternal age and migration background) to 58.1% (item about hospital admission in the first year of an infant’s life).

Thus, 1937 women (mean age, 30.5 ± 4.1 years; range, 17–44 years) were included in the study, of whom 88.1% did not smoke (*n* = 1706), 10.8% used tobacco cigarettes (*n* = 209), 0.5% used e-cigarettes (*n* = 10), and 0.6% were dual users (*n* = 12). In total, 13.1% of non-smokers had quit smoking in the year before their pregnancy. Women who smoked tobacco cigarettes or were dual users had smoked for more than 3 years before their current pregnancies, 85.7% and 58.3% respectively. Of the e-cigarette users, 40.0% had used the device for more than 3 years before their pregnancies.

Table 1 summarizes the demographic and lifestyle characteristics by smoking status. Of note, most women were considered Western (99.0%) and had a partner (97.7%).

### 3.2. E-Cigarette Users

Women who used e-cigarettes were older (typical age ≥ 31 years) than women in the other three subgroups (typical age < 31 years). They also had higher education levels than tobacco users, whereas dual users had similar education levels to the tobacco users (Table 1). All e-cigarette users had a partner, while tobacco users and dual users were more often single. Compared with non-smokers, we found that smokers (tobacco, e-cigarette, and dual users) more often had exposure to second-hand smoking. The nicotine dose in the e-cigarettes also varied between e-cigarette and dual users. Among the ten e-cigarette users, three used variants containing no nicotine (30.0%) and seven used variants containing a low dose (70.0%), while among the twelve dual users, three used no nicotine (25.0%), six used a low dose (50.0%), and three used a high dose (25.0%).

### 3.3. Maternal and Infant Characteristics and Outcomes

Table 2 and Table 3 summarize the characteristics and outcomes of mothers and infants by maternal smoking status. The e-cigarette and dual users were more often primiparous compared with the other subgroups. Women who used e-cigarettes during pregnancy had a much higher proportion of miscarriages before 20 weeks of gestation (30%) compared with either non-smokers (4.1%) or tobacco cigarette smokers (5.7%). Furthermore, compared with non-smokers, women who smoked (tobacco, e-cigarettes, and dual users) more often gave birth to SGA infants.

## 4. Discussion

This study describes the characteristics of Dutch women who used e-cigarettes in pregnancy compared with non-smokers, tobacco cigarette users, and dual users. In our sample, women who used e-cigarettes were older than non-smokers and tobacco users and had more often completed higher education than tobacco users. Compared with the other groups, e-cigarette users were also more likely to have a partner and to have been exposed to second-hand smoke. Moreover, they were more likely to be primiparous, have a pregnancy that ended in a miscarriage, and give birth to SGA infants.

The findings that e-cigarette users were older and had higher education levels compared with tobacco cigarette users, dual users, or non-smokers may reflect the age at which different women have their first infant. E-cigarette users were more often primiparous, possibly reflecting their education levels or age, with higher-educated women tending to be older first-time mothers [30]. A study among young adults in New York City reported that adolescents with higher education levels were more likely to smoke e-cigarettes [31]. By contrast, a study in the UK showed that women with lower education used e-cigarettes more often [26]. The difference in findings could be due to contrasting policies around e-cigarette use during pregnancy in the UK and the Netherlands. In the UK, e-cigarette use is preferred to tobacco cigarette use in pregnancy, whereas both practices are discouraged in the Netherlands [27,32]. Interestingly, e-cigarette users were also more often primiparous in our sample. A Norwegian study reported that women who smoked tobacco cigarettes during their first pregnancy had more often quit smoking before their second pregnancy, leading to a lower prevalence of smoking among multiparous women [33]. This might apply to e-cigarette users too, potentially explaining our finding of relatively more primiparous than multiparous e-cigarette users. We also found that all e-cigarette users in our study had a partner. Studies of tobacco use have shown higher levels among single women, consistent with the high number of single and dual tobacco users in this research [33,34].

When looking at pregnancy outcomes, the pregnancies of women who used e-cigarettes ended more often in a miscarriage compared with the other groups. This result has not been reported in the literature and may represent an incidental finding due to the low number of e-cigarette users. However, tobacco cigarette use during pregnancy has been associated with miscarriage, and our findings indicate that this might also apply to e-cigarette users [35]. However, we are aware that the subgroup of e-cigarette users is small; therefore, no firm conclusions can be drawn.

Furthermore, women in all three smoking groups had higher proportions of SGA infants than non-smokers, consistent with existing literature [24,36]. When investigating the effect of e-cigarette use on birth weight, Cardenas et al. found that users had a higher chance of having an SGA newborn [24]. Our study adds that the characteristics of e-cigarette and dual users (i.e., typically older and primiparous) may account for this association. Further research on the association between e-cigarettes and pregnancy outcomes should account for these characteristics. Finally, the composite adverse maternal outcomes and composite adverse neonatal outcomes showed no statistical differences between the women based on their smoking status. This is not consistent with research on tobacco smoking and adverse pregnancy outcomes [1,2,3,4,6,7,8]. One explanation for this could be the woman’s previous smoking. Research has shown that smoking during pregnancy is associated with gestational hypertension [37]. Because 13.1% of non-smokers smoked in the year preceding their pregnancy, this, along with other factors such as age and parity, may have confounded the effect of current smoking on maternal and infant outcomes. As policy support, future research with a larger sample size should look into the effect of e-cigarette and tobacco use on pregnancy outcomes.

### Strengths and Limitations

To our knowledge, this is one of the first studies to describe the characteristics of women based on their smoking status, including e-cigarette users. To prevent recall bias, we recruited women who had been pregnant for a maximum of 1 year before completing the questionnaire.

An important characteristic that was included in this study is previous smoking behavior, as the effects of previous smoking can have a lasting effect on pregnancies, even after smoking cessation [37]. A limitation of the study is that we did not inquire about information on cessation support for the non-smokers, either with or without nicotine replacement therapy. Among the various limitations of this work, the mostly online recruitment could have introduced selection bias, favoring responses from women interested in participation [38]. To reduce this bias, we also recruited women through parent-child centers, which most Dutch infants and their parents attend in the first 4 years after birth [39]. However, despite the varied recruitment strategies, we only included a relatively small group of e-cigarette users. In our study, 0.5% of women used an e-cigarette during pregnancy, which increased to 1.1% when including dual users. Given that previous data in the Netherlands indicated that only 0.4% of pregnant women reportedly used substances like hookah, nitrous oxide, and/or e-cigarettes, this could represent either selection bias or a true growth in e-cigarette use [17]. Large differences between sample and population can arise by chance in small samples, which most statistical tests will not capture. The low power of the study precluded the use of multivariate analyses. Therefore, more extensive research into e-cigarettes and their associations with pregnancy outcomes is warranted, with, for example, data from biological samples (e.g., urinary cotinine to indicate nicotine exposure) or nationwide cohort data from medical records [40]. Currently, data from Dutch maternity care records is collected in the Perined database [41]. Though the Perined database is very valuable for research, data on the detailed smoking status of pregnant women is not available in this database yet.

## 5. Conclusions

Women who used e-cigarettes during pregnancy were on average older, had higher education levels, were more often primiparous, and more often had miscarriages and SGA infants compared with non-smokers, tobacco cigarette smokers, and dual users. These findings, coupled with the existing literature linking tobacco cigarettes to adverse pregnancy outcomes, should pave the way for more extensive research into e-cigarettes and their effects on pregnancy outcomes. This could be facilitated by midwives and obstetricians improving the data they record on smoking status in electronic health registries. Until more is known, Dutch practitioners should continue to follow existing guidelines and not recommend e-cigarette use during pregnancy [27].

## Figures and Tables

**Figure 1 ijerph-20-02632-f001:**
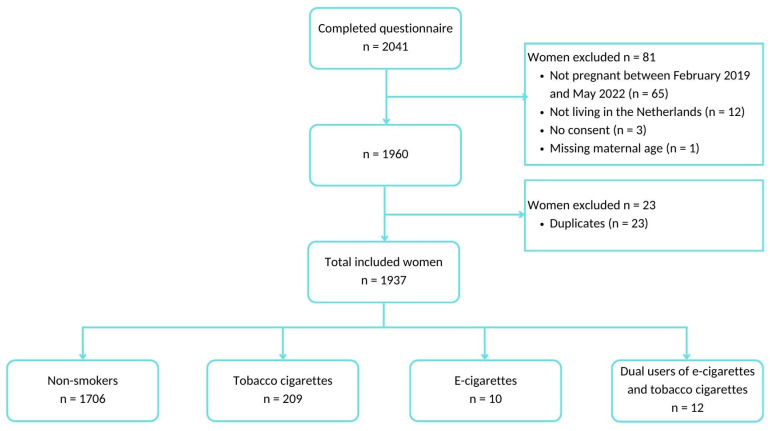
Flowchart of the included population.

**Table 1 ijerph-20-02632-t001:** Demographic and lifestyle characteristics by smoking status (N = 1937).

		Total Population	Maternal Smoking Status				*p* *
			Non-Smokers	Cigarette Users			
				Tobacco	e-cig	Both	
		N (%)	N (%)	N (%)	N (%)	N (%)	
		1937 (100%)	1706 (88.1%)	209 (10.8%)	10 (0.5%)	12 (0.6%)	
Maternal age							**≤0.001**
	18–30 years	998 (51.5)	857 (50.2)	128 (61.2)	2 (20)	11 (91.7)	
	31–35 years	728 (37.6)	663 (38.9)	60 (28.7)	4 (40)	1 (8.3)	
	36–40 years	180 (9.3)	156 (9.1)	20 (9.6)	4 (40)	-	
	≥40 years	31 (1.6)	30 (1.8)	1 (0.5)	-	-	
Migration background ^a^							
	Western	1918 (99.0)	1688 (98.9)	208 (99.5)	10 (100)	12 (100.0)	0.77
	Non-Western	19 (1.0)	18 (1.1)	1 (0.5)	-	-	
Education level ^b^							**≤0.001**
	Low	129 (6.7)	94 (5.5)	32 (15.3)	-	3 (25.0)	
	Middle	845 (43.6)	709 (41.6)	124 (59.3)	6 (60)	6 (50.0)	
	High	941 (48.6)	892 (52.3)	42 (20.1)	4 (40)	3 (25.0)	
	Missing	22 (1.1)	11 (0.6)	11 (5.3)	-	-	
Marital status							**≤0.001**
	Partner	1893 (97.7)	1677 (98.3)	198 (94.7)	10 (100)	8 (66.7)	
	Single ^c^	44 (2.3)	29 (1.7)	11 (5.3)	-	4 (33.3)	
Second-hand smoke							**≤0.001**
	Not exposed	262 (13.5)	261 (15.3)	-	1 (10)	-	
	Exposed	588 (30.4)	485 (28.4)	90 (43.1)	7 (70)	6 (50.0)	
	Missing	1089 (56.1)	960 (56.3)	119 (56.9)	2 (20)	6 (50.0)	
Smoked throughout pregnancy							**≤0.001**
	Not smoked	1706 (88.1)	1706 (100.0)	NA ^d^	NA	NA	
	Part of pregnancy	110 (5.7)	NA	100 (47.8)	5 (50)	5 (41.7)	
	Full pregnancy	121 (6.2)	NA	109 (52.2)	5 (50)	7 (58.3)	
BMI start pregnancy							0.50
	Not obese	980 (50.6)	859 (50.4)	107 (51.2)	7 (70)	7 (58.3)	
	Obese	844 (43.6)	752 (44.1)	84 (40.2)	3 (30)	5 (41.7)	
	Missing	113 (5.8)	95 (5.6)	18 (8.6)	-	-	

* Statistical differences among the four smoking statuses. *p*-value in bold if less than alpha 0.05. ^a^ Western background = birth in Austria, Belgium, Canada, Croatia, Czech Republic, Denmark, Estonia, Finland, France, Germany, Hungary, Iceland, Ireland, Italy, Latvia, Lichtenstein, Lithuania, Luxembourg, Malta, Monaco, Norway, Poland, Portugal, San Marino, Slovakia, Slovenia, Spain, Sweden, The Netherlands, United Kingdom, United States, or Vatican City [29]. ^b^ Education level = low (none or primary school), middle (secondary school), high (higher education). ^c^ Single = divorced, widowed, single, or not married (not cohabiting). ^d^ NA = not applicable e.g., non-smokers have not smoked for any part of their pregnancy.

**Table 2 ijerph-20-02632-t002:** Maternal characteristics and outcomes by smoking status.

		Total Population	Maternal Smoking Status				*p* *
			Non-Smokers	Cigarette Users			
				Tobacco	e-cig	Both	
		N (%)	N (%)	N (%)	N (%)	N (%)	
		1937 (100%)	1706 (88.1%)	209 (10.8%)	10 (0.5%)	12 (0.6%)	
CHARACTERISTICS							
Conception							0.15
	Spontaneous	1774 (91.6)	1555 (91.1)	198 (94.7)	9 (90)	12 (100.0)	
	Artificial reproductive treatment ^a^	160 (8.3)	149 (8.7)	10 (4.8)	1 (10)	-	
	Missing	3 (0.2)	2 (0.1)	1 (0.5)	-	-	
Parity							**0.01**
	Primipara	870 (44.9)	761 (44.6)	92 (44.0)	7 (70)	10 (83.3)	
	Multipara	1017 (52.5)	906 (53.1)	107 (51.2)	2 (20)	2 (16.7)	
	Missing	50 (2.6)	39 (2.3)	10 (4.8)	1 (10)	-	
Mode of birth							0.27
	Spontaneous vaginal birth	1458 (75.3)	1291 (75.7)	154 (73.7)	3 (30)	10 (83.3)	
	Assisted vaginal birth	130 (6.7)	116 (6.8)	12 (5.7)	1 (10)	1 (8.3)	
	Cesarean section	257 (13.3)	224 (13.1)	29 (13.9)	3 (30)	1 (8.3)	
	Missing	92 (4.7)	75 (4.4)	14 (6.7)	3 (30)	-	
ADVERSE OUTCOMES							
Hypertensive disorders							0.71
	No	1748 (90.2)	1540 (90.3)	189 (90.4)	9 (90)	10 (83.3)	
	Yes	189 (9.8)	166 (9.7)	20 (9.6)	1 (10)	2 (16.7)	
Gestational diabetes							0.15
	No	1821 (94.0)	1608 (94.3)	193 (92.3)	8 (80)	12 (100.0)	
	Yes	116 (6.0)	98 (5.7)	16 (7.7)	2 (20)	-	
Postpartum hemorrhage							0.18
	No	1803 (93.1)	1585 (92.9)	199 (95.2)	9 (90)	10 (83.3)	
	Yes	134 (6.9)	121 (7.1)	10 (4.8)	1 (1)	2 (16.7)	
Composite adverse maternal outcome ^b^							0.27
	No	1533 (79.1)	1351 (79.2)	168 (80.4)	6 (60)	8 (66.7)	
	Yes	404 (20.9)	355 (20.8)	41 (19.6)	4 (40)	4 (33.3)	
Miscarriage							**0.01**
	No	1852 (95.6)	1636 (95.9)	197 (94.3)	7 (70)	12 (100.0)	
	Yes	85 (4.4)	70 (4.1)	12 (5.7)	3 (30)	-	

* Statistical differences among the four smoking statuses; *p*-value in bold if less than alpha 0.05. ^a^ Artificial reproductive treatment: conception through in-vitro fertilization, intra-cytoplasmatic sperm injection, intra-uterine insemination, or donor. ^b^ Maternal adverse outcome: hypertensive disorder, gestational diabetes, or postpartum hemorrhage; or any combination of the three.

**Table 3 ijerph-20-02632-t003:** Infant characteristics and outcomes by smoking status.

		Total Population	Maternal Smoking Status				*p* *
			Non-Smokers	Cigarette Users			
				Tobacco	e-cig	Both	
		N (%)	N (%)	N (%)	N (%)	N (%)	
		1937 (100%)	1706 (88.1%)	209 (10.8%)	10 (0.5%)	12 (0.6%)	
CHARACTERISTICS							
Sex							0.88
	Female	917 (47.3)	809 (47.4)	100 (47.8)	3 (30)	5 (41.7)	
	Male	929 (48.0)	823 (48.2)	95 (45.5)	4 (40)	7 (58.3)	
	Missing	91 (4.7)	74 (4.3)	14 (6.7)	3 (30)	-	
Gestational age							0.75
	Preterm birth (<37 weeks)	86 (4.4)	76 (4.5)	9 (4.3)	-	1 (8.3)	
	Term/post-term birth (≥37 weeks)	1761 (90.9)	1557 (91.3)	186 (89.0)	7 (70)	11 (91.7)	
	Missing	90 (4.6)	73 (4.3)	14 (6.7)	3 (30)		
ADVERSE OUTCOMES							
Size gestational age ^a^							**0.02**
	Small for gestational age	95 (4.9)	77 (4.5)	15 (7.2)	2 (20)	1 (8.3)	
	Missing	117 (6.0)	97 (5.7)	17 (8.1)	3 (30)	-	
Hospital admission first life year							0.43
	No	687 (35.5)	604 (35.4)	73 (34.9)	4 (40)	6 (50.0)	
	Yes	121 (6.2)	108 (6.3)	11 (5.3)	2 (20)	-	
	Missing	1129 (58.3)	994 (58.3)	125 (59.8)	4 (40)	6 (50.0)	
Composite adverse infant outcome ^b^							0.11
	No	1649 (85.1)	1464 (85.8)	170 (81.3)	5 (50)	10 (83.3)	
	Yes	173 (8.9)	147 (8.6)	22 (10.5)	2 (20)	2 (16.7)	
	Missing	115 (6.0)	95 (5.6)	17 (8.1)	3 (30)	-	

* Statistical differences among the four smoking statuses; *p*-value in bold if less than alpha 0.05. ^a^ Size at gestational age, as reported by women and defined as “the baby was too small for the gestational age.” ^b^ Infant adverse outcome: preterm birth, small for gestational age, hospital admission, or stillbirth; or any combination of the three.

## Data Availability

The data is not publicly available.

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
