# Peer review of "Adverse Maternal and Infant Outcomes of Women Who Differ in Smoking Status: E-Cigarette and Tobacco Cigarette Users"

_ijerph, 2023, doi:10.3390/ijerph20032632_

Round 1

Reviewer 1 Report

This paper focuses on a very important topic of today and should encourage further research. I have a few comments for the authors to take into consideration.

It is unclear whether you collected data on previous smoking. This data is also valuable and may lead to biased results. Please clarify this in methods/results section and consider discussing about it and the influences it may have. Also persons on replacement nicotine therapy, where they excluded? This would have maybe explained why e-cig users were older as this may have been persons that swithched at some point.

You mention Figure 1 in text, however it is missing. Please revise.

There is a lot of missing data. It should be described in the Methods section how this data was handled.

Your study is a cross-sectional study, not a cohort so please correct this in the discussion. Although, a cohort would be of great value and might implicate some causality.

Discussion should focus a bit on outcomes also. Now it seems you were just describing the characteristics of pregnant smokers.

Can all references be placed at the end of sentence please, it is hard to follow as it is now.

Reviewer 2 Report

Thank you for giving me the opportunity to read this well written research report on the characteristics of pregnant women and the maternal and infant outcomes by smoking status. 

I only have one minor query: Line159 indicated that there should be Figure 1? Figure 1 is not included in the submission. 

Reviewer 3 Report

- line 57, please explain EVALI

- at what stage of pregnancy/ post-pregnancy did women answer the questionnaire? This is unclear. Also, was questionnaire designed so that women had to answer all questions? How much data was missing ? 

- did women get compensation for filling out QA ? 

- Table 2 - it appears there is a mistake in the line multiparra, should be 2 and not 20 ? (numbers don't add up to 10)

Round 2

Reviewer 3 Report

Please discuss limitations of your study in the discussion section. 

There was no collection of data from biological samples (urinary cotinine) or data on infant and maternal health from hospital records.

There were only 10 women in the study reporting e-cig use, which makes the  conclusions on health effects not convincing.

The authors found that almost 10.8% of women smoke conventional cigarettes in pregnancy but don't comment on this finding. The authors don't address the risk reduction argument, perhaps shift to e-cigarettes in this 10.8% of the population might reduce maternal and infant harm COMPARED to conventional cigarettes. 
